# HPV Involvement in the Tumor Microenvironment and Immune Treatment in Head and Neck Squamous Cell Carcinomas

**DOI:** 10.3390/cancers12051060

**Published:** 2020-04-25

**Authors:** Jérôme R. Lechien, Géraldine Descamps, Imelda Seminerio, Sonia Furgiuele, Didier Dequanter, Francois Mouawad, Cécile Badoual, Fabrice Journe, Sven Saussez

**Affiliations:** 1Department of Otolaryngology and Head and Neck Surgery, CHU Saint-Pierre, 1000 Brussels, Belgium; jerome.lechien@umons.ac.be (J.R.L.); Didier.Dequanter@pandora.be (D.D.); 2Department of Otolaryngology and Head and Neck Surgery, CHU of Lille, University Lille 2, 59000 Lille, France; Francois.Mouawad@chru-lille.fr; 3Department of Human Anatomy and Experimental Oncology, Faculty of Medicine, Research Institute for Health Sciences and Technology, University of Mons (UMONS), Avenue du Champ de Mars, 8, B7000 Mons, Belgium; Geraldine.Descamps@umons.ac.be (G.D.); Imelda.Seminerio@umons.ac.be (I.S.); Sonia.Furgiuele@umons.ac.be (S.F.); Fabrice.Journe@umons.ac.be (F.J.); 4Department of anatomo-pathology, G Pompidou European Hospital, AP-HP, University of Paris, 75015 Paris, France; cecile.badoual@aphp.fr; 5Laboratory of Oncology and Experimental Surgery, Institute Jules Bordet, Free University of Brussels, Rue Heger-Bordet, 1, B1000 Brussels, Belgium

**Keywords:** HPV, cancer, head and neck, immunity, microenvironment, treatment

## Abstract

Head and neck squamous cell carcinomas (HNSCC) are one of the most prevalent cancers worldwide. Active human papillomavirus (HPV) infection has been identified as an important additional risk factor and seems to be associated with a better prognosis in non-drinker and non-smoker young patients with oropharyngeal SCC. The better response of the immune system against the HPV-induced HNSCC is suspected as a potential explanation for the better prognosis of young patients. To further assess this hypothesis, our review aims to shed light the current knowledge about the impact of HPV infection on the immune response in the context of HNSCC, focusing on the innate immune system, particularly highlighting the role of macrophages, Langerhans and myeloid cells, and on the adaptative immune system, pointing out the involvement of T regulatory, T CD8 and T CD4 lymphocytes. In addition, we also review the preventive (HPV vaccines) and therapeutic (checkpoint inhibitors) strategies against HPV-related HNSCC, stressing the use of anti-CTLA4, PD-L1, PD-L2 antibodies alone and in combination with other agents able to modulate immune responses.

## 1. Introduction

Head and neck squamous cell carcinomas (HNSCC) have been intensively studied regarding their high incidence worldwide, ranking fifth among most frequent cancers in men and eleventh in women [1,2]. Based on the updated data base Globocan, more than 880,000 new HNSCC have been identified in both sexes with a men/women ratio of 3:1 in 2018 [3]. In the last years, many studies conducted in the US observed a decrease incidence of head and neck cancers among men and women populations concomitantly with a significant increase of oropharyngeal cancers, specifically in men [4]. The majority of HNSCCs are traditionally related to tobacco and alcohol exposure, but the high-risk human papillomavirus infection is now recognized as a causal agent responsible for the development of a subset of oropharyngeal carcinomas [5,6]. Although some differences in terms of prevalence are reported between Europe, US and Asia, the trends of HPV-related HNSCC continue to rise significantly, unlike to the non-HPV related HNSCC which has declined for 30 years [7]. HPV infection is predominantly assigned to types HPV-16 and HPV-18, but geographical heterogeneities of 60% in the US and 31% in Europe have been reported in recent studies [8]. HNSCC can be classified into two groups, depending to their risk factors: firstly, HNSCC occurring in young people (≤40 years old) who never smoke/drink, which is usually associated with HPV infection; and, secondly, HNSCC associated with smoking and drinking habits occurring in non-infected older people (>40 years old) [9,10]. This classification makes sense according to the HNSCC prognosis. Indeed, although these cancers are often associated with a poor prognosis, because they are discovered in advanced stages (stages III–IVB), the presence of these risk factors seems to have a significant impact on patient’s prognosis. Thus, some clinical studies suggested three conditions that affect the prognosis differently: first, smokers/drinkers without HPV infection which manifest a poor overall survival, secondly, HPV-positive patients without history of alcohol and tobacco intoxication which present the better outcome, and thirdly, an intermediate overall survival in HPV+ patients consuming alcohol and/or tobacco [11,12,13,14,15,16,17]. Besides the overall survival, one of the major endpoints evaluated regarding HPV infection is the loco-regional tumor (LCR) control. Indeed, the HPV status appeared many times as a strong prognostic factor of LCR after primary and postoperative radiochemotherapy of locally advanced HNSCC [18,19]. The differences between these groups could be due to the HPV infection and the related interactions with the immune system. The aim of this review is to summarize the current knowledge about the impact of HPV infection on immune system and immune treatment in HNSCC. Precisely, we review experimental and clinical published data on activities of stromal and immune cells in HPV-related head and neck cancers.

## 2. Human Papillomavirus

HPVs are small circular double-stranded DNA viruses (±8000 base pairs), non-enveloped, that can infect epithelial cells especially those present in the upper-aerodigestive tract. Those viruses can be classified into cutaneous (responsible for genital warts) or mucosal types, mucosal HPV types being mostly found in potentially malignant and cancerous lesions of the epithelium, leading to their classification as “high-risk HPVs” (HPV-16, -18, -31, -33, -34, -35, - 39, -45, -51, -52, -56, -58, -59, -66, -68, -70) [20,21].

In HNSCC, HPV-16 remains the most prevalent type, with more than 90% of HPV-associated HNSCC linked to HPV-16 [6,22,23]. HPV-16 DNA coding sequences are classified into early (E) genes (E1, E2, E4, E5, E6 and E7) and late (L) genes, coding for the major (L1) and minor (L2) proteins of the viral capsid. The structure and role of HPV proteins have been fully reviewed by Rautava & Syrjänen and Doorbar et al. in 2012 [20,21]. In cancer cells, HPV DNA can be either in episomal form (not integrated in the host cell genome), integrated form (its DNA is integrated into the host cell genome) or both. Particularly in high-risk HPV types, the deregulated and elevated expression of E6 and E7 from high-grade dysplasia generates many genetic disorders in the infected cell, facilitating the integration of viral episomes into the host cell DNA. The integration occurs in most cases through E2, resulting in the loss of E6/E7 regulation which lead consequently to cancer. Indeed, E6 and E7 proteins are able to disrupt important regulatory pathways in the cell, such as those mediated by the retinoblastoma protein family, and by p53, which controls functions such as cell cycle entry, differentiation and proliferation [24,25]. In more detail, E6 recruits the ubiquitin-ligase E6AP, leading to the poly-ubiquitination and the degradation of p53 [20,26,27]. E7 binds to the retinoblasma protein (pRb), which is in hypo-phoshorylated form, bound to the transcription factor E2F. By inducing the phosphorylation and degradation of pRb, E7 leads to the separation of pRb/E2F complex, releasing E2F. The inhibition of pRb allows the overexpression of p16, which tries to block cell cycle by inhibiting cyclin D/CDK4-6 complex [20,28]. This overexpression of p16 is widely used as a clinical biomarker of HPV infection, particularly for oropharyngeal SCC, and is commonly employed in many clinical trials [29]. However, its accuracy for other head and neck subsites is controversial, because p16 overexpression can be induced by HPV-independent mechanisms, and a significant number of discordant cases (p16+ IHC/PCR or ISH-) are also now described in the literature. Recently, Lechner has suggested a relation between a deregulation of chromatin state by inactivating mutations in NSD1 and high expression of p16 in HPV-non-oropharyngeal carcinomas [30]. Although the detection of RNA transcripts remains the most accurate technique to evaluate HPV infection, a combined strategy evaluating p16 by immunohistochemistry and HPV DNA by PCR (GP5+/GP6+) has demonstrated valuable efficiency with similar sensibility and specificity rates [31,32,33]. Depending on E7 activity, p16 upregulation is now accepted as a marker of active HPV involvement in tumor cells, so HPV/p16+ tumors have a transcriptionally active HPV infection, while HPV/p16− tumors present the viral DNA but without viral oncogene expression [30].

HPV infection is mostly a sexually transmitted disease. It can occur during sexual intercourse, when mechanical abrasions allow the virus to squame to the basal layer of the epithelium. Autoinoculation is also possible between an infected and a non-infected area of the patient’s body [34,35]. Vertical transmission from mother to child may also occur during pregnancy (transplacental) or delivery, this type of transmission being higher during vaginal delivery and when the mother is infected with several types of HPVs [36,37,38]. In HNSCC, HPV infection happens through orogenital contacts, and is four times higher in men, especially given their higher number of sexual partners, compared to women [34,39].

## 3. Immune System and HPV Infection

Most women infected with a high-risk HPV have an elimination of their infection, which is mediated by the immune system, especially after recognition of viral antigens by T lymphocytes. However, it is estimated that the immune system of 10% to 15% of infected women is not reactive enough to eliminate the virus, leading to persistent infection [25]. Therefore, chronic lesions occur, allowing HPV to stagnate at the top of the epithelium for months to years. Host immune responses that appear contribute to virus maintenance, HPV being able to adapt in order to escape immune monitoring. Indeed, HPV remaining in epithelial cells is able to make blind the immune system by limiting the expression of its genes at very low levels (10–100 copies per cell), sharply decreasing the presentation of viral antigens on MHC class I, and therefore recognition by the innate immune system. Adaptive responses are also decreased, especially by restricting the maintenance of Langerhans cells in infected epithelia [40,41,42,43]. In fact, HPV infection does not induce cell lysis, but rather a release of virions towards the apical pole of the epithelium, which does not allow Langerhans cells to present viral antigens [44]. Indeed, the reaction of the immune system depends on the ability of antigen-presenting cells to detect HPV and to present its viral antigens to competent immune cells. It has been showed that the most recognized antigens in low-grade lesions are from E2 and E6 proteins, while epitopes from E7 oncoprotein induce the biggest response in high-grade lesions [45,46,47].

Recently, the modulation of the immune signature has been highlighted regarding the HPV integration status. This study revealed a significantly elevated expression of genes specifically expressed in T-cells (CD4+, Regulatory, CD3+, and CD8+), NK cells, and B cells in integration-negative tumors, with this group of tumors with non-integrated HPV being associated with a better outcome compared to integrated-tumors [48].

The immune-modulatory role of high-risk HPV proteins has also been demonstrated through the downregulation of NF-κB pathway to evade the immune system, especially through the binding of E6 and E7 with some coactivators of NF-κB in the nucleus, resulting in persistence of HPV infection [12,49,50,51].

The virus can also keep the immune system blind by disrupting secreted cytokines, allowing immune escape. Moreover, HPV decreases the sensibility of infected cells to immune system by deregulating the type 1 IFN response via E6 and E7 oncoproteins [52]. The virus early protein E5 has also been demonstrated to be responsible for MHC class I CD1d downregulation, leading to escape to Natural Killer cells [53]. The decrease of viral antigens specific T lymphocytes, and the increase of regulatory T lymphocytes and anti-inflammatory cytokines (IL-10, TGFβ) defeat the immune reaction, preventing HPV clearance [41,44]. This loss of control of HPV proliferation and of viral oncogenes expression leads to a change in the phenotype of the lesion that evolves towards dysplasia and finally to cancer [54].

## 4. Innate Immune System and HPV-Related HNSCC

### 4.1. Macrophages

Macrophages constitute strong mediators of innate and adaptive inflammatory responses, particularly in the fight against cancer [55]. Macrophages arise from monocytes of the bone marrow, and they circulate in the bloodstream and penetrate organs where there are needed. Some monocytes differentiate into resident macrophages in the organs, such as Kupffer cells in the liver, osteoclast in bones, macrophages in pulmonary alveoli, microglia in the brain, etc [56]. The latest macrophage nomenclature distinguished two main macrophage subsets, which undergo a polarization process in response to microenvironmental stimuli: M1 macrophages, also named “classically activated macrophages”, present a pro-inflammatory activity and are involved in Th1 type of responses, and M2 macrophages referenced as “alternatively differentiated macrophages”, produce anti-inflammatory cytokines. M1 can be stimulated by IFNγ, lipopolysaccharide (LPS) and the granulocyte-macrophage colony-stimulating factor (GM-CSF), which generate the production of pro-inflammatory cytokines, such as interleukins IL1β, IL6, IL12, IL18, IL23, and the tumor necrosis factor (TNFα). Their phenotypic profile is characterized by the expression of MHCII, CD68, CD80, and CD86 [57]. The main characteristics of M2 macrophages are as follows: they are induced by CSF-1, IL4, IL10, IL13, and the transforming growth factor β (TGF-β) with the consequence to produce high concentrations of IL10. Phenotypically, M2 express particularly the macrophage mannose receptor (MMR), corresponding to CD206, as well as CD200R, CD163, MGL1, MGL2 [58]. However, it has been accepted for many years that macrophages possess a huge plasticity and diversity, and that this binary classification will be an outdated concept [59]. Indeed, recent studies proposed that polarized macrophages M1 and M2 will be the extremes of a continuum of macrophage polarization [60]. In addition, a model has been developed suggesting the existence of a spectrum of differentiated macrophages. However, the wealth of information about the different macrophages categories and phenotypes are yet to be fully understood and discovered. Moreover, it is important to note that several categories of macrophages not defined as M1 or M2 exist. These include the existence of tumor-associated macrophages (TAMs), CD169+ macrophages, TCRαβ+ and TCRγδ+ macrophages, which are involved in immune tolerance, inflammatory and infectious disease [57,58,60]. 

TAMs are derived from monocytes attracted in the tumor microenvironment (TME) (via CCL2, CCL8) and are differentiated trough IL-4, IL-10, IL-13 and TNF-α stimuli [57,61]. These macrophages are only present in tumors and share some M1 and M2 characteristics. TAMs are usually called “M2-like macrophages”, because they share some similarities: same cytokines stimuli, CD163 expression, IL-10 and CCL2 production [58]. TAMs play essential roles in tumorigenesis. They are implicated in angiogenesis (VEGF, TGF-β, MMPs secretion), in migration and invasion (MMPs, EGF, serine proteases secretion), in epithelial to mesenchymal transition (EMT) (TGF-β secretion), in intravasation and extravasation (CCL18 chemokine production), in the interaction with cancer stem cells and finally they are implicated in immunosuppression (PD-L1/PD-L2 expression, IL-10, TGF-β, arginase-1 and prostaglandins production) [57]. 

Regarding macrophages behavior in a tumoral microenvironment, M1 present anti-tumoral effects: they contribute to cytotoxic CD8+ T cell activation and naïve CD4+ T cell differentiation into Th1 effector cells [62,63,64]. M2 macrophages exhibit pro-tumoral actions like TAMs, which are consider as pro-tumorigenic immune cells. They stimulate regulatory T cell differentiation and secrete several factors (e.g., TGFβ, TNFα and IL-10) to create a favorable environment for tumor progression and to inhibit the anti-tumor effects of immune cells (Figure 1) [65,66]. 

In most studies, the main marker to stain the whole population of macrophages (both M1 and M2 macrophages) is CD68 [67,68,69,70]. Regarding their prognostic value, several studies showed that TAMs are associated with tumor progression and poor overall survival in cervical cancers [71,72,73] and other HPV-related cancers [74]. Moreover, a high recruitment of macrophages is observed in metastatic oral squamous cell carcinomas (OSCC) and contributes to lymph node metastasis and poor survival [75,76]. Our recent study has shown a high infiltration of these immune cells, which was also correlated with shorter recurrence-free and overall survival of the patients, demonstrating that the macrophages number is an independent prognostic factor for HNSCC patients [77] (Figure 2). 

We have also demonstrated that CD68+ macrophages were strongly recruited during the tumor progression from the peri-tumoral tumor free epithelia until dysplasia and carcinomas (Figure 3). Furthermore, when we have looked at the M1/M2 ratio in the tumor micro-environment, it appears that more than 80% of stained macrophages are macrophages of the M2 phenotype, namely TAMs [75,78]. In cervical cancers, patients with this high ratio have longer disease-free survival and present a more complete response to chemoradiation [79]. It seems clear that the polarization of macrophages into TAMs and their infiltration in the tumor micro-environment is a poor prognostic factor. Indeed, high levels of TAMs are associated with poor prognosis in several HPV+/p16+-related cancers [80,81,82,83]. In addition, TAMs are correlated with poor overall survival and poor clinical outcomes in oral carcinomas [84,85,86], given that they increase the invasion, migration and, angiogenesis [87,88,89]. In HNSCC, high levels of TAMs in the tumor micro-environment are correlated with poor prognosis, because of the CTLA-4-mediated immunosuppression and the expression of immunosuppressive cytokines and PD-L1 [77,90]. HNSCC cells interact with monocytes and induce their polarization into TAMs, which secretes EGF and TGFβ, promoting the EMT of cancer cells. [86]. TAMs can also decrease the functional activity of T cells by expressing Arg-1 and iNOS, responsible for L-arginine depletion, a precursor of their metabolism [74]. 

Finally, by secreting IL-10, TAMs promote the differentiation of T lymphocytes into regulatory T lymphocytes (Figure 1) [91]. Bellmunt and colleagues demonstrated that macrophages foster the laryngeal cancer cell migration thanks to exosome signaling. Moreover, exosomes also induce the expression of IL-10 in macrophages and PD-L1 in cancer cells, so resulting in the promotion of an immunosuppressive environment. They showed that the effects induced in cancer cells are mediated by the exosome-depending activation of STAT-3 signal transduction pathway [92]. In 20% to 25% of HNSCC, patients have HPV infection. The comparison of HPV-negative tumors versus HPV+/p16+ tumors in our recent study showed that the recruitment of macrophages was the highest in HPV+/p16+ patients, when studying both the intra-tumoral and the stromal compartments [77]. HPV acts as an immunosuppressor by decreasing the polarization and activation of macrophages in M1 macrophages, and by increasing the secretion of TGFβ, leading to the activation of TAMs [93,94]. However, less is still known about the impact of HPV on the recruitment of TAMs in HNSCC.

### 4.2. Langerhans Cells

Only three studies examined the involvement of Langerhans cells in the development of HNSCC regarding the HPV infection. In 1996, Li et al. found that HPV infection was associated with a reduction of the number of Langerhans cells and the related destruction of the immune surveillance system, favoring the development of esophageal carcinomas [95]. These authors found a higher number of Langerhans cells in both intratumoral and stromal compartments of HPV− tumors compared to HPV+ tumors. Fifteen years later, Pereira et al. did not find significant differences about the stromal infiltration of Langerhans cells between HPV+ and HPV− oral SCC samples [96]. These authors suggested that the HPV infection does not alter the immunological system and the Langerhans infiltration in oral SCC. The last paper focused on the infiltration of Langerhans cells throughout the carcinogenesis in 125 samples of HPV+ and HPV− HNSCC [97]. First, these authors found that Langerhans cell infiltration increases throughout the carcinogenesis (from normal tissue and low dysplasia to severe dysplasia and carcinoma) (Figure 3) but decreases with HPV infection. Second, the intraepithelial number of Langerhans cells was higher in HPV+/p16− tumors than in HPV+/p16+ tumors but, in the stromal compartment, there was no significant difference in Langerhans cell infiltration between HPV+/p16+ and HPV+/p16− HNSCC. Moreover, stromal over-infiltration of Langerhans cells was associated with better overall survival and recurrence free-survival only in the HPV− tumor samples (Figure 2), suggesting that the Langerhans cell number is an independent prognostic factor for HNSCC [97]. Most recently, Silva and co-authors observed a decrease of CD1a+ cells when comparing 45 cases of oral SCC to eight samples of normal epithelia [98].

### 4.3. Myeloid Cells

Four studies reported data about the tumor infiltration of myeloid-derived-suppressor cells (MDSCs) in HPV+ and HPV− HNSCC [99,100,101,102]. Partlova et al. exhibited that HPV+ HNSCC present a higher tumor intraepithelial infiltration of MDSCs than HPV− HNSCC. In this study, this specific infiltration, as well as the infiltration of other immune cells (Treg, CD4+, CD8+ T cells), was associated with a higher cytokine production in the stromal compartment, including CCL-17, 21; IL-10, 17, 21; TNF-α; and IFN-γ. In addition, immune cell infiltration was also associated with an overexpression of PD-1 in HPV+ tumor samples [101]. The increased infiltration of MDSCs in the stromal compartment of HNSCC was also confirmed in the study of Russel et al., but they did not report a significant difference between HPV+ and HPV− HNSCC [100]. Similar findings have been reported by Yu et al., who also did not find substantial differences of stromal MDSC infiltration between HPV+ and HPV− HNSCC [99]. However, these authors identified a significant relationship between the stromal MDSC infiltration and the expression of PD-L1. This study supports that PD-1/PD-L1 expressions and the MDSC population were increased in both HPV+ and HPV− HNSCC. Ma et al. also found a higher MDSC infiltration in tumoral samples of HPV+ oral and oropharyngeal SCC than dysplasia and normal tissues [102]. Recently, Ryan and co-authors demonstrate that STAT1 is an essential mediator of the antitumor response through the inhibition of myeloid derived suppressor cell accumulation and promotion of T-cell mediated immune responses in murine HPV− head and neck squamous cell carcinoma [103]. Many other studies were conducted with the aim to investigate antitumor effects in HPV+ murine models, so Li et al. recently published that their candidate vaccine against HPV16E7 significantly reduced Treg and MDSCs infiltration in mouse genital tumors [104,105]. Given the creation of a new immunocompetent mouse model of HPV16+ head and neck squamous cell carcinoma [106], this promising vaccination strategy should be considered and tested for the treatment of HPV+ oropharyngeal SCC.

Additionally, in order to reduce the immunosuppressive microenvironment of HNSCC, Mao and colleagues focused on phosphorylation levels of SRC and LYN kinases in a HNSCC mouse model treated with Dasatinib. Their results indicated that inhibition of LYN by Dasatinib reduces the population of MDSCs and that LYN overexpression correlated with stromal MDSCs and TAMs, resulting consequently with a poor prognosis of HNSCC patients [107]. These findings indicate that LYN could be a potential target in future immunotherapy treatments. 

Three studies have been conducted on the peripheral blood circulation of MDSCs [102,108,109]. Al-Taei et al. studied the evolution of peripheral blood MDSCs before and after the treatment of oropharyngeal SCC [108]. First, these authors reported that the peripheral blood concentration of MDSCs was significantly higher in patients with oropharyngeal SCC than healthy controls. Second, they found a higher posttreatment MDSC peripheral blood concentration in comparison to pre-treatment, suggesting that HPV-targeted immunotherapy in post-treatment oropharyngeal SCC patients could require multiple strategies to boost T cell immunity, and to overcome the influence of immunosuppressive cells. Parikh et al. also investigated the blood concentration of MDSCs in patients with HPV+/p16+ oropharyngeal SCC treated by chemoradiation [109]. They reported similar findings than Al-Taei et al. regarding the higher number of MDSC blood concentration in patients with oropharyngeal SCC than healthy controls, and, in patients with HPV+/p16+ induced oropharyngeal SCC, they also found a significantly higher MDSC peripheral blood concentration after CRT in comparison with the pretreatment concentration [109]. Finally, Ma et al. reported that the MDSC peripheral blood level was significantly higher in HNSCC than in healthy controls, but this result only concerned HPV+ oral and oropharyngeal SCCs [102]. Overall, HNSCC samples seem to have a higher infiltrate of MDSCs in intraepithelial and/or stromal compartments than dysplasia and normal tissues, but this infiltration does not seem to be related to the HPV status. According to the peripheral blood concentration, it seems that the MDSC blood concentration is higher in patients with HNSCC than healthy controls, and this concentration could increase post-treatment, suggesting a reduction of the HPV-specific T cell responses post-treatment.

## 5. Adaptative Immune System and HPV-Related HNSCC

### 5.1. Regulatory T Lymphocytes

Fifteen researches studied the involvement of Treg in the development of HNSCC according to HPV status. The impact of the HPV infection on Treg infiltration in both intraepithelial and stromal compartments is still debated, because studies reported controversial results. Among the studies comparing the Tumor Infiltrating Lymphocytes (TILs) between HPV+ and HPV− samples, seven studies found significant difference of TILs according to HPV status [100,101,110,111,112,113,114], and four did not find substantial difference [115,116,117,118].

Among these studies, Kindt et al. and Seminerio et al. investigated the infiltration of Foxp3 Treg in both stromal and intraepithelial tissues of HNSCC [111,114]. These authors showed a progressive stromal compartment infiltration throughout the carcinogenesis (Figure 3). In the stromal compartment, the infiltration of Treg did not differ between HPV+ and HPV− tumor samples, while HPV+ tumor samples were characterized by a significant higher number of Treg in the intraepithelial compartment compared to HPV− tumor samples [111]. In a similar study, Russel et al. found a significantly higher infiltration of Foxp3 Treg in the stromal compartment of HPV+ samples compared to HPV− samples but there was not substantial difference in the intraepithelial compartment regarding the HPV status [100]. In another study, Punt et al. found a higher infiltration of Treg in both intraepithelial and stromal compartments of HPV+ oropharyngeal SCC samples in comparison with HPV− tumor samples. In addition, these authors have found that a higher infiltration of Foxp3 Treg in both the intraepithelial and stromal compartments was associated with better overall survival and disease-free survival [112]. Thus, they supported that the infiltration of Foxp3 Treg is a strong prognostic factor in oropharyngeal SCC. In the same way, we recently published that a high infiltration of Foxp3+ Treg in the stromal compartment of head and neck tumors was significantly associated with a worse prognosis in terms of recurrence-free survival (RFS) and overall survival (OS) (Figure 2) [111,114]. The three remaining research studies from Partlova et al., Nasman et al. and Badoual et al. also showed a higher infiltration of Treg in the intraepithelial compartment of HPV+ HNSCC than the HPV− tumor [101,110,113]. Treg infiltration of the stromal compartment was not studied in these three papers. The other studies did not find significant difference of stromal and/or intraepithelial infiltrated lymphocytes between HPV+ and HPV− HNSCC samples [115,116,117,118]. 

Two studies investigated the expression of some important ligands by Treg, according to the HPV status, and demonstrated that Treg infiltration is correlated with an overexpression of B7H4 ligand, PD-L1, and FasL by Treg in HNSCC [100,109] (Figure 1). The relationship between the expression of these ligands and the HPV status seems, however, uncertain and requires additional investigations. 

The involvement of Treg in the carcinogenesis of HPV+ SCC has also been investigated throughout the Treg concentration in peripheral blood of patients with HPV induced SCC. In fact, Treg blood concentration seems to be elevated in many patients with HNSCC, irrespective of HPV status [108,116,119]. Al-Taei et al. and Parikh et al. investigated the peripheral blood concentration of HPV-specific Treg in patients with oropharyngeal SCC treated by chemoradiation regarding HPV status [108,109]. Interestingly, these two studies reported a decrease of Treg at the end of the treatment in comparison with the pretreatment state. These studies mainly strengthen the post-treatment immunosuppression of HPV-specific T cells that underlies the need for multiple strategies to boost T cell immunity, and to overcome the influence of immunosuppressive cells. On the contrary, Masterson et al. found a significant peripheral blood increase in Treg in HPV+/p16+ oropharyngeal SCC after treatment [120]. Heusinkveld et al. and Lukesova et al. also did not find significant differences between HPV+ and HPV− patients in the peripheral blood level of Treg in hypopharyngeal, oropharyngeal, and oral SCC [116,121]. However, it is important to note that all these studies were characterized by a low number of SCC samples, especially those positive for the HPV infection.

### 5.2. T CD8 Lymphocytes 

The contribution of T cells in antitumor activity and in tumor progression is more and more explored and understood, but it seems crucial to clarify their prognostic value and their role in head and neck TME. The adaptative response of the immune system is based on B cells by the humoral response and on T cells by the cell-mediated response. Two classes of T lymphocytes can be distinguished: the cytotoxic T-lymphocytes characterized by the expression of CD8, which target and destroy infected cells when activated, and the CD4+ helper T-lymphocytes, which activate CD8+ T cells, promote the anti-tumor response, and catalyze the humoral immune response. Both play a role in T cell recognition and activation by binding to their respective class I and class II major histocompatibility complex (MHC) ligands on an antigen presenting cell (APC). 

Oropharyngeal tumors are highly immune infiltrated, but conflicting results are reported concerning the relevance of immune infiltration in HPV+ versus HPV− oropharyngeal SCCs. Studies reported either increased immune infiltration in HPV+ tumors [122,123], increased overall survival and disease-free survival in highly infiltrated tumors, irrespective of their HPV-status [118], or no significant difference between HPV+ and HPV− tumors [124]. Recently, Schneider et al. demonstrated a significantly higher infiltration of CD3+ and CD8+ T-lymphocytes in p16 positive oropharyngeal tumors, compared to p16 negative oropharyngeal, laryngeal, hypopharyngeal, and oral cavity carcinomas, but the CD8+ infiltrate did not correlate with patient’s prognosis [125]. Overall, most studies focusing on oropharyngeal SCCs reported a higher CD8+ TILs infiltrate in HPV+ compared to HPV− SCCs [110,115,120,124,126,127,128,129,130,131,132]. The same observation is sometimes observed when authors do not distinguish oropharyngeal from other localizations, so many researchers found a significantly increased infiltration of intratumoral CD8+ TILs in HPV+ HNSCCs compared to HPV− HNSCCs [100,101,133]. Contrariwise, a similar infiltration of T-lymphocytes, including CD8+, was also observed in HPV+ and HPV− HNSCCs [134]. More specifically, these authors demonstrated different pattern of CD8+ T-cell infiltration, according to the tumor localization: the higher CD8+ infiltrate in HPV+ tumors were observed only in the oropharyngeal SCCs [129,135]. 

The prognostic value of CD8+ T cells is frequently assessed. Indeed, a lot of studies assessing the prognostic value of CD8+ TILs found a better outcome for HPV−, as well as HPV+ patients with high CD8+ T cell infiltration. However, all studies focusing on HPV+ oropharyngeal SCCs reported that CD8+ T-cell infiltration was associated with a good prognosis and an improved overall survival and locoregional control [110,115,118,123,126,127,132,136]. A similar observation was made in laryngeal carcinomas, where an elevated CD8+ infiltrate was found to be associated with HPV+ laryngeal SCCs and an improved outcome [137]. Moreover, two consecutive studies of de Meulenaere et al. evaluated the relation between TILs and outcome. Particularly, an elevated CD8+ T cell count was significantly associated with prolonged OS versus patients with a low CD8+ T cell count. They proved that solely CD8+ infiltrating T cells exhibit a positive effect on OS [138,139]. In conclusion, it seems that CD8+ T lymphocytes infiltrate constitutes an independent prognostic marker in patients diagnosed with oropharyngeal squamous cell carcinoma, but it is not reliable enough for other localizations.

### 5.3. T CD4 Lymphocytes

Only a few publications reported data about CD4+ infiltrate in the HPV-related head and neck carcinogenesis. Among these, most observed a significantly higher number of CD4+ T-cells in HPV+ oropharyngeal SCCs compared to HPV− ones [124,126,127,128]. However, they did not find any relation to patient outcomes. Regarding the prognostic value, the role of CD4+ T-cells remains unclear because it is associated with controversial results. It was demonstrated by van Kempen et al. that HPV-positive oropharyngeal SCCs were significantly more heavily infiltrated by TILs, notably CD4+ and CD8+, compared to HPV-negative OPSCCs, and that this high level of TILs was correlated to a good prognosis in HPV+ OSCCs [131]. Oguejiofor et al., reported that a high CD3+CD4+ TILs infiltration was not statistically correlated with an improved overall survival [115]. Similar observations were found in the studies of Balpermpas et al. and Lee et al., where CD4+ expression was not related to the overall survival rate [134,140]. Regarding publications in HPV− patient cohorts, a high CD4+ TILs infiltrate was associated with a better overall survival, as well as with a better locoregional control [115,130,131,134]. Despite the number of studies examining the CD4 count and their prognostic value, the role of CD4+ lymphocytes remains ambiguous. Due to this lack of certainty, another parameter is frequently analyzed: the CD4/CD8 ratio. A better outcome was notably observed in patients with a low CD4/CD8 ratio in the tonsils and at the base of the tongue carcinomas, but this was regardless of HPV status [118]. On the contrary, Wolf et al showed that CD4/CD8 ratio of infiltrating cells tended to be higher in patients with better overall survival but was not statistically significant [76]. Recently, Zhang et al. evaluated the effect of HPV status on the CD4/CD8 ratio in laryngeal SCCs, and the survival benefit and found that this ratio of TILs was lower in HPV+ patients. Moreover, they did not find significant correlations with progression-free survival [137]. Furthermore, a study focusing on advanced laryngeal cancers demonstrated that increased peripheral blood CD4 levels were highly predictive of a response to induction chemotherapy and a trend toward improved overall survival [141]. Considering the ambiguous role of CD4+ T-lymphocytes in head and neck TME, we conclude that more studies need to be done to understand and to confirm the prognostic role of CD4.

## 6. Preventive and Therapeutic Approaches to HPV-Related HNSCC

### 6.1. Human Papillomavirus Vaccines

To date, there is no specific antiviral treatment against HPV infections, with most of these infections (up to 90%) being asymptomatic and resorbing spontaneously within 2 years. For this reason, priority is given to screening and, in the case of lesions in the cervix, to the treatment of pre-invasive cervical lesions, particularly via burn ablation, cryotherapy or surgical excision [142]. 

Vaccination is the only way to prevent HPV infection, because it is done at the time of puberty (from the age of 9), so before the beginning of sexual life and the exposure to the virus. Three prophylactic vaccines are currently available worldwilde: Gardasil^®^, Cervarix^®^ and Garadasil-9^®^, both using L1 virus-like particles (VLP), which will generate neutralizing antibodies against HPV major capsid protein L1. Gardasil^®^ (Merck & Co., Whitehouse Station, NJ, USA), the first L1 VLP recombinant vaccine approved by the US Food and Drug Administration (FDA) in 2006, protects against two low-risk HPV subtypes -6 and -11, and against two high-risk HPV types -16 and -18. This vaccine is produced by using a yeast substrate and amorphous aluminium hydroxyphosphate sulfate (AAHS) as adjuvant. Each dose of Gardasil^®^ vaccine contains 20 µg, 40 µg, 40 µg and 20 µg of L1 protein of HPV-6, -11, -16 and -18 respectively, adsorbed on 225 µg of AAHS [143]. Cervarix^®^ (GlaxoSmithKline Biologicals, Rixensart, Belgium), has been introduced in vaccination campaigns in 2007 and contains L1 VLP allowing the protection against high-risk HPV-16 and -18. It is produced in Trichoplusia ni cells by using a baculovirus expression system and contains an adjuvant system composed of aluminium hydroxide and 3-O-desacyl-4-monophosphoryl lipid A (AS04). Each dose of this vaccine contains 20 µg of L1 protein of HPV-16 and 20 µg of L1 protein of HPV-18 adsorbed on 500 µg of aluminum hydroxide and 50 µg of AS04. Finally, Gardasil 9^®^ (Merck & Co., Whitehouse Station, NJ, USA) has been approved by the FDA in 2014, and is similar to the quadrivalent vaccine Gardasil^®^, except that vaccination extends to HPV types -31, -33, -45, -52 and -58. Each dose of Gardasil 9^®^ contains 30 µg, 40 µg, 60 µg, 40 µg of L1 protein of HPV6, -11, -16 and -18, respectively and, 20 µg of each L1 protein of HPV types -31, -33, -45, -52 and -58, adsorbed on 500 µg of AAHS [144]. 

These three prophylactic vaccines are indicated for girls and boys from 9 years old, in order to protect against pre-malignant anogenital lesions occurring in cervical, vulvar, vaginal and anal regions. The quadrivalent vaccine protects also against anogenital condyloma and furthermore the nonavalent vaccine protects against genital warts. The action of these vaccines relies mainly on the stimulation of antibody-mediated immunity, allowing the detection of HPV infection before the virus infects basal cells of the epithelium. The efficacy of HPV vaccines has been deeply demonstrated by conducting several extensive studies in girls and boys worldwide, but this vaccination is unfortunately not extended to boys in all countries. Indeed, in the United States, the FDA approved Gardasil^®^ for the vaccination of girls from 9 to 26 years old in 2006, and of boys between 9 and 21 years old in 2011. In Europe, most of the countries introduced HPV vaccination campaigns for girls, but this is generally not the case for boys. For example, Belgium started HPV vaccination in 2008 for girls but the decision to extend this vaccination to boys has only been taken by the government in 2018. However, extensive clinical studies have been conducted on Gardasil^®^ and Cervarix^®^, and have strongly demonstrated the efficacy (up to 100%) of both of them against cervical, vulvar and vaginal neoplasia, cervical cancer, anal cancer and genital warts, both in females and males [145,146,147]. The quadrivalent vaccine Gardasil^®^ also protects against anal intraepithelial neoplasia and genital warts [148,149,150]. Finally, recent studies have suggested that HPV vaccination might also protect against oral HPV infections, both in females and males [148,151,152], suggesting its use as a protection against oropharyngeal carcinomas [153,154]. Indeed, a clinical study conducted in Costa Rica on 7466 aged 18–25 years old women, who randomly received Cervarix^®^ or a Hepatitis A vaccine as a control, showed that HPV16/18 oral prevalence was decreased in HPV-vaccinated women compared to the control population, estimating the efficacy of Cervarix^®^ at 93.3%. The authors have hypothesized that this vaccine might provide protection against HPV16/18, with potential implications in the prevention against HPV-related oropharyngeal carcinomas [155].

### 6.2. Immune Checkpoint Inhibitors

Recent understanding of the complex interactions between HNSCC and immune cells supported the valuable role of immunomodulating agents in the therapeutic strategies including surgery, radiotherapy and chemotherapy. To obtain a significant effect of immune system, interactions between immune and cancer cells are needed to trigger a strong immune response [156]. Indeed, tumor infiltrating lymphocytes (TILs) are associated with a favorable clinical outcome in HNSCC patients [157]. However, alteration of HLA class I molecule expression [158], increase of cytokines (IL-6, TGFβ) [159,160], as well as the activation of STAT3 or NF-κB [160,161] have been reported as potent immunosuppressive mechanisms involved in HNSCC (Figure 1).

A critical point of attention with the activation of T-cell responses is to achieve a balance between co-stimulatory and co-inhibitory agents [162]. Hence, the discovery of co-inhibitory receptors such as Cytotoxic T-Lymphocyte-associated Antigen 4 (CTLA-4) or Programmed Death-1 (PD-1) brought new possibilities to activate immune response against tumor cells using monoclonal antibodies to such co-inhibitory immune checkpoints. In several types of cancer, the receptor CTLA-4 is expressed by activated T cells, where it binds to B7 and prevents its interaction with CD28, consequently decreasing the regulation, leading to a negative regulation of T cell proliferation and IL-2 production. As expected, blockade of CTLA-4 using an anti-CTLA-4 monoclonal antibody (Ipilimumab) correlates with an activation of T-cells in metastatic melanoma patients [163,164]. 

Therefore, the inhibition of such interactions increased T cell response and mediated anti-tumor activity. In 2016, FDA approved the Programmed cell Death-1 (PD-1) monoclonal antibodies Nivolumab (Opdivo^®^, Bristol-Myer Squibb, US) and Pembrolizumab (Keytruda^®^, Merck & Co., Inc., US) for the treatment of HNSCC. These anti-PD-1 monoclonal antibodies are designed to block co-inhibitory signaling mediated by the PD-1/PD-L1 axis. Nivolumab improves both progression-free and overall survivals in HNSCC patients, with a durable benefit for responders (but only 30% of patients are durable responders) [165]. Pembrolizumab is used for patients with metastatic HNSCC and is demonstrated to improve PFS and OS [166]. Concerning PD-1 ligands, PD-L1 is considered as a validated biomarker, and is used to predict the response to immune checkpoint inhibitor. However, the robustness of this biomarker is sometimes not enough. There is a turnover in the expression of PD-L1 protein, which can explain this gap [167]. PD-L2 is another and less characterized PD-1 ligand expressed in HNSCC, which could be also used as a predictive biomarker. An immune checkpoint formation, PD-1/PD-L1 or PD-1/PD-L2, decreases cytokine production and induces T lymphocyte apoptosis, leading to cancer cells immunevasion. In parallel, the evaluation of PD-L1 inhibitors, such as the monoclonal antibodies Durvalumab, Atezolizumab and Avelumab, is currently ongoing in HNSCC [168].

However, recent evidences reported the activation of alternative immune checkpoints in patients with resistance to PD-1 or PD-L1 antibodies [169], supporting the use of combination of checkpoint inhibitors to overcome such resistance and improve response rates and patient survival. In this context, the dual blockade of PD-1 and CTLA-4 is currently explored in many cancers such as melanoma, where this combination improves the response rate [170], and four independent phase I/II, II and III trials are under evaluation for the combination of PD-1 and CTLA-4 blockade in HNSCC (U111-1166-0687, NCT02823574, NCT02741570, NCT02551159).

In addition, several other agents able to modulate immune responses are currently investigated for the treatment of HNSCC patients. Such trials include the combination of anti-PD-1 antibody with a STAT3 inhibitor or CXCR2 antagonist in recurrent/metastatic HNSCC (NCT02499328). In this context, the indoleamine 2,3-dioxygenase 1 (IDO1), activated by inflammation, catabolizes tryptophan to kynurenine with the consequence to induce immune tolerance by suppressing T-cells. IDO has been found to predict poor clinical outcome in laryngeal squamous cell carcinoma patients [171]. Then, the results of a phase I/II study evaluating the combination of Pembrolizumab with an oral inhibitor of IDO1 in patients with recurrent/metastatic HNSCC suggest promising anti-tumor activity and weak side effects [172].

Additionally, several oncolytic viruses are currently evaluated in HNSCC, as they can reduce tumor burden and stimulate anti-tumor immunity. Currently, an ongoing clinical study is considering the combination of Talimogene Laherparepvec (T-VEC), with pembrolizumab in recurrent/metastatic HNSCC (NCT02626000). Based on this therapeutic combinatory option, HPV could be view as an “endogenous virus”, able to stimulate immunity, explaining the better outcome of patients with HPV-associated cancers, as well as the better response to immunotherapies [51]. In fact, HPV+ HNSCC upregulates PD-L1 and PD-L2 on fibroblast via TLR9. Moreover, macrophages stimulate PD-L1 and PD-L2 expression on HPV+ HNSCC. This induces an immunosuppressive tumor environment [173]. Like we have already explained in this review, HPV plays a critical role in TME modulation, and should also be considered as a biomarker for immunotherapy administration [41].

In parallel to the development of immunomodulatory drugs, many studies are searching for biomarkers of response prediction, attempting to select patients and to assist clinician in monitoring response [174]. Based on the fact that HPV status should be considered, both Nivolumab and Pembrolizumab studies have explored the impact of HPV on clinical outcome. These studies demonstrated that p16+ patients treated with Nivolumab had significantly the highest survivals compared to p16− ones [166,175,176]. Hence, p16 could predict stronger response to immunotherapy. Moreover, the benefit obtained in patients with PD-L1+ cancer has also been observed in these trials, suggesting the potential of this ligand as a promising biomarker [166,175]. In HNSCC, the combinations of p16 and PD-L1 may give additional predictive value to patient selection for checkpoint inhibitor therapy.

### 6.3. Impact of Metformin on the Tumor Microenvironment

Another interesting point to highlight is the importance of metformin on the tumor immune microenvironment modulation. Metformin, which is known for its antidiabetic properties, has also revealed some interest in carcinogenesis. This drug can modulate the expression of innate, as well as adaptive, immune cells in the TME. 

The expression of HIF-1α by tumor cells under hypoxic conditions is a well-known mechanism, and its role in tumor progression and metastasis is more and more characterized [177]. Metformin has been shown to reduce HIF-1α mRNA and protein expressions in HNSCC cell lines. Moreover, metformin decreases HSP-90, which leads to no increase of bcl-2, essential to stabilize HIF-1α [178]. Recently, Amin et al., have demonstrated in a clinical study of 36 HNSCC patients, HPV+ patients (*n* = 16) versus HPV− patients (*n* = 20), that metformin induces a decrease of LTreg Foxp3+ in the intratumoral compartment, and increases the number of LTCD8+ cells in the stromal compartment, respectively, in both populations. CD8+/Foxp3+ ratio is higher after metformin treatment in both the tumor (*p* < 0.001) and stromal (*p* < 0.001) compartments. However, no significance differences were observed between HPV status [179]. Concerning macrophages, hypoxia promotes their polarization in M2-phenotype in HNSCC cells by VEGF and IL-6 secretion. Then, M2-type TAMs express CCL15, which binds to CCR1 on HNSCC and activates the NF-κB pathway. Interestingly, Metformin breaks down the interaction between M2 and HNSCC via CCL15-CCR1 induced by hypoxia. This mechanism leads to the inactivation of the NF-κB pathway and consequently to the apoptosis of the cell [180]. In this context, a new study reinforced previous observations, where metformin modulates the TME metabolism, which impacts immune cells and inhibits the growth and the proliferation of cancer cells [181]. It seems obvious that metformin generates an interest in the field of combined therapies, because of its impact on PD-L1 and many actors of the TME [182]. 

## 7. Conclusions

The better understanding of the role of immune system in cancer is currently revolutionizing the management of such disease. It brings rationale for the prognostic and the treatment of HNSCC, especially for the combination of therapeutic strategies, in parallel to the development of predictive biomarkers for personalized medicine. In this context, many data, including ours, indicated that HPV, mainly evidenced by p16 immunodetection, is involved in the recruitment of immune cells within the tumor. It is important to note that caution should be used when assessing p16 status alone as a marker of active HPV infection, because p16 overexpression may be also due to alterations of signaling pathways in cancer. In many studies, p16 status predicts a favorable patient outcome in oropharyngeal cancer. However, conflicting studies did not reveal differences between HPV-positive and negative tumors regarding patient survival, mainly because of the difference in the assessment methods of HPV and/or p16 status, the small size of the population, as well as the lack of accurate in anatomic locations of the tumors. To definitively address this question, we are currently evaluating p16 (nuclear and cytosol staining in more than 70% of the tumor) and overall survival in a cohort of 700 patients with various anatomical sites of HNSCC. 

It is now well-documented that HPV infection plays a role in the recruitment of immune cells within the HNSCC. However, the mechanisms by which HPV-tumor cells favor infiltration of specific immune cells bring some inconsistent findings, because of the incomplete evaluation of both intra-tumoral and stromal compartments, the lack of specificity/sensitivity of used antibodies, as well as the absence of a clear consortium defining the antigens to target for a specific immune cell. Indeed, immune response is a very dynamic process, requiring the assessment of all subpopulations of immune cells, in order to understand their interplay and the consequences on the patient prognosis. In this context, panels of antibodies should be used for evaluating tumor-associated macrophages (TAMs), tumor-infiltrating lymphocytes (TILs) and cytokines to further complete our understanding of immune response facing to HPV-related HNSCC.

With the approval by the FDA of PD-1checkpoint inhibitors (Nivolumab and Pembrolizumab) to stimulate immune response, such therapeutic strategy has already offered new treatment options in recurrent/metastatic HNSCC. Through many aspects, “combinations” appear to be the current step to study and clinically exploit such approach, evaluating p16 expression and immune cell recruitment for the prognosis, as well as p16 and PD-L1 levels for the prediction of response to immunotherapies. However, the rapid turnover of proteins, such as PD-L1 may make them not fully useable for response prediction. Therefore, the two main challenges to improve the benefits of immunotherapy in HNSCC are the discovery of reliable predictive biomarkers for the selection of patients who should benefit from immune checkpoint inhibitors, and the optimization of the combination of the most effective treatments modulating the immune system to overcome resistance associated to monotherapy, and to improve response rate as well as patient survival. The improving knowledge of TME in the different subgroups of head and neck cancer patients emphasizes the need for the development of combinations of immunological biomarkers, recently named “immunoscore”, in order to bring prognostic information and to guide clinicians toward the most appropriate decision for patients. 

## Figures and Tables

**Figure 1 cancers-12-01060-f001:**
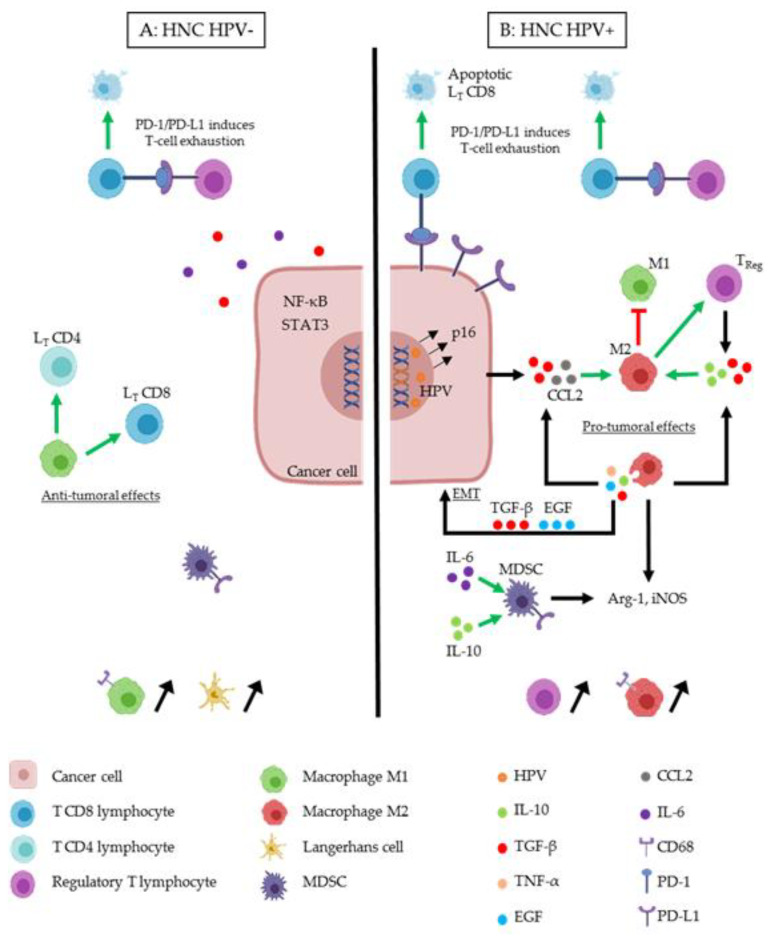
Immune cells interactions in the tumor microenvironment of head and neck cancer (HNC) HPV− (**A**) and HNC HPV+ (**B**). Cancer cells can modify the tumor microenvironment (TME), in order to induce immunosuppression. Several actors play an essential role. (A) M1 macrophages (green cell) induce anti-tumoral effects by activating LT CD4 (light blue cell) and LT CD8 (dark blue cell). In the TME of HNC HPV-, IL-6 (purple round) and TGF-β (red round) are secreted in order to induce immunosuppression like the expression of transcription factors (NF-κB and STAT3). TReg cells (purple cell) act as an immunosuppressive cell by inducing T-cell exhaustion. TReg express PD-L1 which binds to its receptor, PD-1, and induces LT CD8 apoptosis, this phenomenon appears also in HPV+ TME (B). M1 macrophages and Langerhans cells (yellow cell) are more present in the TME of HNC HPV-. (B) HPV (orange circle) infect HNC cell and integrate the host cell genome. This induces the expression of oncoproteins and the overexpression of p16. M2 macrophages (red cell) are predominant in this TME. TGF-β and CCL2 (grey circle) secreted by the HPV+ cancer cell act to differentiate macrophages in the M2 phenotype, which acts as a pro-tumoral immune actor. M2 inhibit M1 and stimulate TReg cells. M2 secrete TGF-β, IL-10 (green round) and IL-6, which induce a feedback on its stimulation, but also TGF-β and EGF (blue round), which induce epithelial to mesenchymal transition (EMT) on cancer cells. IL-6, and IL-10 stimulate myeloid-derived-suppressor cells (MDSC) cell (dark cell), which secrete Arg-1 and iNOS like M2 macrophages.

**Figure 2 cancers-12-01060-f002:**
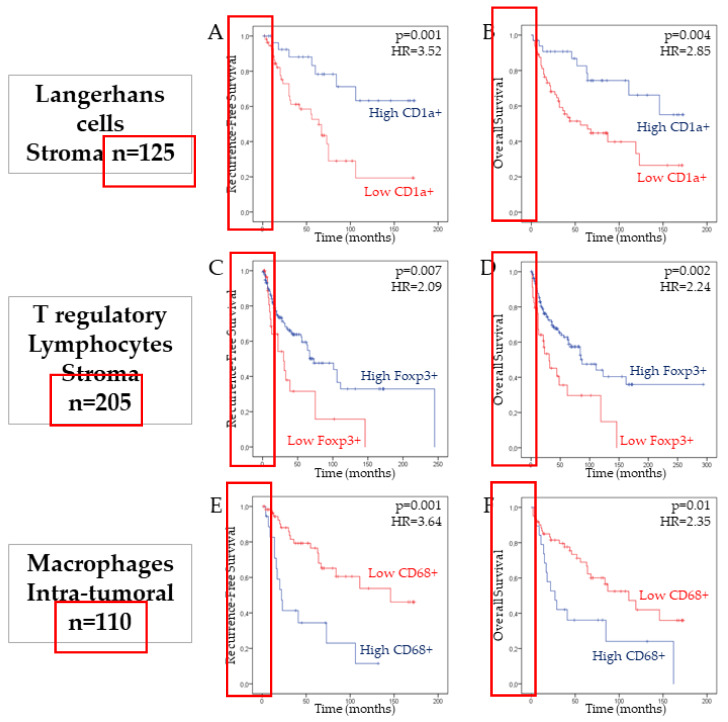
Association between CD1a+ Langerhans cells, FoxP3+ Treg and CD68+ macrophages infiltration and patient survival in head and neck squamous cell carcinomas (HNSCC). Kaplan–Meier curves of the recurrence-free survival (RFS) (**A**,**C**,**E**) and overall survival (OS) (**B**,**D**,**F**) of patients with HNSCC in the stromal compartment, according to the number of CD1a+ Langerhans cells (*p* = 0.001 and *p* = 0.004, log-rank test) and Foxp3+ Treg cells (*p* = 0.007 and *p* = 0.002, log-rank test). The intra-tumoral number of macrophages is associated with a lower RFS (*p* = 0.001) and OS (*p* = 0.01, log-rank test) of HNSCC patients.

**Figure 3 cancers-12-01060-f003:**
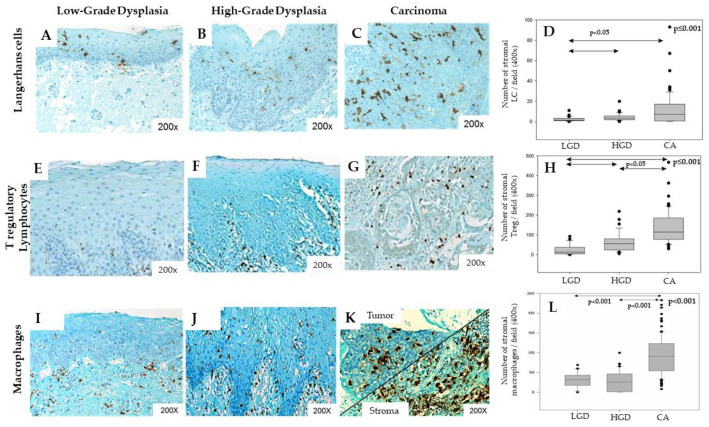
Langerhans cells, Treg lymphocytes and macrophages infiltration increases during HNSCC progression. Immunohistochemical representation of CD1a+ Langerhans cells, FoxP3+ Treg and CD68+ macrophages in Low-Grade Dysplasia (LGD) (**A**,**E**,**I**), High-Grade Dysplasia (**B**,**F**,**J**), and carcinoma (CA) (**C**,**G**,**K**) from head and neck patients. Kruskall–Wallis tests illustrating the increasing number of Langerhans cells (**D**, *p* ≤ 0.001), Treg lymphocytes (**H**, *p* ≤ 0.001) and macrophages (**L**, *p* < 0.001) in the stromal compartment during tumor progression.

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
