# Peer review of "HPV Involvement in the Tumor Microenvironment and Immune Treatment in Head and Neck Squamous Cell Carcinomas"

_cancers, 2020, doi:10.3390/cancers12051060_

Round 1

Reviewer 1 Report

Summary

The Review article by Lechien et al. is well written and provides a thorough body of work discussing a topic relevant to the readers of Cancers and the broader community of head and neck cancer researchers. However, some points should be addressed prior to publication and are also noted as comments within the PDF document:

Major comments

  1. Consider revising the abstract and throughout the review article text when making associations between HPV status and p16 IHC. It is important to note that a significant number of discordant cases (ie. HPV neg HNSCC that are p16 IHC pos., HPV ISH neg.) are now described in the literature so caution should be used when assessing p16 status alone with respect to TIME: 
    1. Lechner M, Chakravarthy AR, Walter V, et al. Frequent HPV-independent p16/INK4A overexpression in head and neck cancer. Oral Oncol. 2018;83:32-37. doi:10.1016/j.oraloncology.2018.06.006.
    2. Prigge E-S, Arbyn M, Von Knebel Doeberitz M, Reuschenbach M. Diagnostic accuracy of p16INK4aimmunohistochemistry in oropharyngeal squamous cell carcinomas: A systematic review and meta-analysis. International Journal of Cancer. 2017;140(5):1186-98.
    3. Rooper LM, Gandhi M, Bishop JA, Westra WH. RNA in-situ hybridization is a practical and effective method for determining HPV status of oropharyngeal squamous cell carcinoma including discordant cases that are p16 positive by immunohistochemistry but HPV negative by DNA in-situ hybridization. Oral Oncology. 2016;55:11-6.

  1. Please clarify the statement “A clear general upward trend has been observed over time, with an increase in cases among women but a decrease in the men population, primarily in the United States, during the last 20 years” further as these trends are confounded by HPV status. For example, an “upward trend” is seen in the US but occurs after 2005 and is primarily due to the increase in the number of HPV+ OPSCC cases and not HNSCC overall. Making this distinction more clearly is important given the focus of the review article on HPV+ HNSCC, and that HPV- HNSCC cases (alcohol and tobacco-related) are on the decline.

Also, a decrease in cases among male men in the US is not apparent according to the recent SEER Cancer Statistics Review (CSR) 1975-2015:

         https://seer.cancer.gov/archive/csr/1975_2015/

Rather, there appears to be an upward trend among males and females after 2005.

Also, I don’t see the decrease in male cases that they mention and again wonder if it has something to do with a specific subset of HNSCC (ex. alcohol and tobacco-related HPV- HNSCC) or perhaps US vs. European stats. Let me know what you think and if I’ve overlooked something here.

  1. Consider citing/referencing more contemporary statistics from papers published on global and US-based populations since the GLOBOCAN statistics are published every 5-8 years. According to the EU registry, it appears that the overall trends are somewhat similar to the US (doi: 10.1371/journal.pone.0192621) – with an important distinction being that the only exception is a lower prevalence of HPV pos HNSCC. Thus, interpretations based on global and/or US data along with GLOBOCAN2018 are recommended.

  1. Binary classification of macrophage phenotypes is now considered an oversimplification as M1 and M2 phenotypes are consider the 2 extremes of a spectrum of phenotypes and does not include Regulatory Macrophages (Mregs). Consider further clarification of this point.

Minor comments

  1. Correct minor grammatical and spelling errors.

Author Response

Reply to the reviewers

Dear reviewer,

We would like to thank you for reviewing our paper and providing relevant comments that will help us to improve the quality of our publication. You will find below the detailing list of our corrections made in our article (in Track Changes) in response to your recommendations.

Major comments

1) Consider revising the abstract and throughout the review article text when making associations between HPV status and p16 IHC. It is important to note that a significant number of discordant cases (ie. HPV neg HNSCC that are p16 IHC pos., HPV ISH neg.) are now described in the literature so caution should be used when assessing p16 status alone with respect to TIME: 

The abstract has been completely revised and rewritten.

Line 106 to 122: an important clarification was also brought concerning the p16 status and its detection techniques. As requested, details have been added on the discordant cases existing in the literature (IHC p16+/ISH HPV-) and your suggested publications have been added.

2) Please clarify the statement “A clear general upward trend has been observed over time, with an increase in cases among women but a decrease in the men population, primarily in the United States, during the last 20 years” further as these trends are confounded by HPV status. For example, an “upward trend” is seen in the US but occurs after 2005 and is primarily due to the increase in the number of HPV+ OPSCC cases and not HNSCC overall. Making this distinction more clearly is important given the focus of the review article on HPV+ HNSCC, and that HPV- HNSCC cases (alcohol and tobacco-related) are on the decline.

The literature has been revised and actualized so we removed this sentence and wrote a new paragraph (lines 44 to 56) pointing the difference between men and women trends regarding HPV status. So, a clear distinction has been made between HPV+ OPSCC and HNSCC.

Also, a decrease in cases among male men in the US is not apparent according to the recent SEER Cancer Statistics Review (CSR) 1975-2015

We agree with your comments so that we corrected it in the new paragraph lines 44 to 56

Rather, there appears to be an upward trend among males and females after 2005.

Also, I don’t see the decrease in male cases that they mention and again wonder if it has something to do with a specific subset of HNSCC (ex. alcohol and tobacco-related HPV- HNSCC) or perhaps US vs. European stats. Let me know what you think and if I’ve overlooked something here.

As explained above, we replaced this paragraph by new data based on more recent studies.

3) Consider citing/referencing more contemporary statistics from papers published on global and US-based populations since the GLOBOCAN statistics are published every 5-8 years. According to the EU registry, it appears that the overall trends are somewhat similar to the US (doi: 10.1371/journal.pone.0192621) – with an important distinction being that the only exception is a lower prevalence of HPV pos HNSCC. Thus, interpretations based on global and/or US data along with GLOBOCAN2018 are recommended.

We totally agree with this comment so that we reviewed prevalence and trends with the Globocan 2018 statistics and modified them at lines 44 to 48.

4) Binary classification of macrophage phenotypes is now considered an oversimplification as M1 and M2 phenotypes are consider the 2 extremes of a spectrum of phenotypes and does not include Regulatory Macrophages (Mregs). Consider further clarification of this point.

Lines 174 to 196: our previous analysis was actually too simplified that is why we have rewritten a new description of macrophages classification with more explanations regarding their different phenotypes and insisting on their diversity and plasticity.

Minor comments

1) Correct minor grammatical and spelling errors.

The article has been completely reread in order to avoid grammatical and spelling errors.

Comments within the PDF document

  • Regarding your suggested publications, they have been all included into the text.
  • Lines 71-74: we also commented on the association between HPV status and locoregional tumor control
  • Lines 106-122: we wrote a paragraph regarding p16 overexpression and techniques of detection
  • In the point 3 “Immune system and HPV infection”, we updated the literature and we added some information at lines 149 to 157
  • Further clarifications have been written regarding TAMs at lines 197 to 206
  • Lines 321 to 332: we revised the paragraph and added details about the creation of a new immunocompetent HPV+ murine model
  • In the point 6.2 “Immune checkpoint inhibitors” (line 520), we developed the role of PD-L2 and we rewrote the paragraph to reduce overlaps with published works.
  • As a point 6.3 (line 609), we discussed about the impact of metformin on HNSCC's tumor immune microenvironment depending on HPV status. So, we added a paragraph entitled: “6.3 Impact of metformin on the TME”

Reviewer 2 Report

Nicely done review of the tumor microenvironment in HPV-related squamous cell carcinoma.

Author Response

Reply to the reviewers

Dear reviewer,

We would like to thank you for reviewing our paper and providing relevant comments that will help us to improve the quality of our publication. You will find below the detailing list of our corrections made in our article (in Track Changes) in response to your recommendations.

We thank you for your appreciation.

Reviewer 3 Report

This review covers background on HPV in HNSCC, studies of immune cells and cytokines in the tumor microenvironment for HNSCC, differences between HPV positive and negative tumors, and the implications. These immune cells include macrophages, Langerhans cells, different classes of T cells, and other myeloid cells, which in some studies showed involvement in HNSCC tumors in general or specifically in HPV positive HNSCC tumors. These and other results were posed as supporting the concept that the enhanced immune environment contributes to HPV positive HNSSC patients exhibiting a better clinical outcome than HPV negative patients. Also reviewed were studies of HPV vaccines in prevention and immuno-modulating drugs as treatments for HNSCC.

In many cases, this review provides useful and pertinent discussion of studies of the immune microenvironment of HNSCC tumors and as they relate to HPV status. The authors make clear that in many cases there are conflicting studies as to whether there are differences between HPV positive and negative HNSCC tumors and other inconsistent findings. This can make the case for results as being inconclusive. The authors choose to lean toward concluding that there are differences in the Conclusion, without stating and or making a case for this leaning. The Conclusion should make clear what is and isn’t in agreement and possibly soften conclusions in which the relevant studies are not completely consistent.

The Abstract does not reflect what is in the review very well. Less on HPV and more on the immune microenvironment would be good.

There are cases in which statements, information, interpretations and conclusions are taken from other reviews that are cited. This can be problematic especially when the cited review gets its information from a previous review such that misinformation, misinterpretations, and misconceptions are perpetuated. Also some reviews are rather old such that as concepts change, they are not updated. So it’s recommended to cite the most recent and original publications.

There are some issues in the description of HPV background listed below.

The use of the term “latent” for only episomal HPV and not integrated HPV doesn’t make sense; they are both latent.

Line 71, “When HPV genome is integrated, the virus main oncoproteins E6 and E7 induce deregulations in host cell cycle in order to promote its progression and proliferation. “ This sentence is misleading since HPV E6 and E7 deregulate p53 and pRb pathways regardless of the integrated or episomal state of the HPV genome. It was theorized in early studies of HPV that integration of the HPV genome resulted in increased expression of E6 and E7 and in turn cancer progression, but data is still needed that would support this theory.   Also, it’s not clear to what ‘its progression and proliferation’ refers to. Presumably, it refers to host cells but ‘host cells’ are not mentioned in the sentence.

Line 78, “This overexpression of p16 is a well established biological marker of HPV infection and is now referenced as a gold standard of active HPV infection.” High p16 protein expression is a marker for HNSCC being HPV positive not for active HPV infections.

Line 80, “HPV/p16+ tumors have a transcriptionally active HPV infection where the virus is in its integrated form and induces the transcription of its genes while HPV/p16 - tumors have a latent HPV infection meaning there is no transcriptional activity for HPV genes.” This sentence has many problems.  All or virtually all HPV positive HNSCC cells have transcriptionally active HPV genomes whether p16 positive or not or independently whether episomal or integrated. It makes no sense to say episomal HPV positive cells have no transcriptionally active HPV genes when it’s established that expression of E6 and E7 is driving the cancer and E1 and E2 expression is required for the autonomous replication of the HPV episome. In fact, most studies that address this topic show that integration of the HPV genome into the host genome disrupts many HPV genes from being expressed as a result of splitting a gene and or dislocation of the primary HPV promoter from the E1, E2, E4 and E5 genes. While studies have shown a correlation between episomal/integrated states of HPV with the level of p16 expression, p16 overexpression has been used as a marker regardless of the HPV genome state though it’s significantly less than 100% accurate.

There are some grammar and word-use issues, some of which are listed below. However, this list is not comprehensive and even the Abstract has at least one issue.

Line 49, “Thus, some clinical studies suggested that patients with a history of smoking and drinking abuses but without HPV infection would have a poorer overall survival than those with HPV- positive tumor and no history of alcohol and tobacco intoxication, and intermediate overall survival in patients with alcohol and smoke abuse and HPV- positive tumor.” This sentence is too messy to understand clearly.

Line 53, the term “would” makes the sentence awkward; “could” works better.

Line 75, the term “binded” should be “bound”.

Line 281, the term “researches” is uncommon and awkward; “research studies” is appropriate.

Line 286, “Two studies interested to the expression of some important ligands by Treg, according to the HPV status.” This sentence is awkward and incomplete and needs restating.

Line 287, the term “would” is awkward in this sentence.

Line 433, “type” should be “types”.

Line 481, the word “rational” does not make sense; the authors may have wanted to use “rationale”.

Author Response

Reply to the reviewers

Dear reviewer,

We would like to thank you for reviewing our paper and providing relevant comments that will help us to improve the quality of our publication. You will find below the detailing list of our corrections made in our article (in Track Changes) in response to your recommendations.

In many cases, this review provides useful and pertinent discussion of studies of the immune microenvironment of HNSCC tumors and as they relate to HPV status. The authors make clear that in many cases there are conflicting studies as to whether there are differences between HPV positive and negative HNSCC tumors and other inconsistent findings. This can make the case for results as being inconclusive. The authors choose to lean toward concluding that there are differences in the Conclusion, without stating and or making a case for this leaning. The Conclusion should make clear what is and isn’t in agreement and possibly soften conclusions in which the relevant studies are not completely consistent.

The Conclusion (line 633) reports now the most relevant information of the review and is commented by the authors about p16 and immune cell recruitment for prognosis and treatment in HNSCC patients. In addition, prospects and ongoing studies are also reported opening new ways of reflection for future studies.

The Abstract does not reflect what is in the review very well. Less on HPV and more on the immune microenvironment would be good.

The Abstract has been completed to better reflect the topics of the review, emphasizing the important roles of immune system and the current therapies against HPV-related HNSCC (lines 24 to 36).

There are cases in which statements, information, interpretations and conclusions are taken from other reviews that are cited. This can be problematic especially when the cited review gets its information from a previous review such that misinformation, misinterpretations, and misconceptions are perpetuated. Also some reviews are rather old such that as concepts change, they are not updated. So it’s recommended to cite the most recent and original publications.

The review has been carefully read to avoid misinformation and misinterpretations. Moreover, many concepts and statements have been updated based on an actualized and recent literature. More recent reviews and studies have been added.

There are some issues in the description of HPV background listed below.

The use of the term “latent” for only episomal HPV and not integrated HPV doesn’t make sense; they are both latent.

To avoid any confusion or misinterpretation, we removed the term “latent” and clarified the paragraph lines 92 to 98.

In response to your comment, we viewed the concept of latency as first, the viral DNA at levels too low to allow life cycle completion or it could be the clearance but persistence with low level viral gene expression with a possible reactivation upon immune depletion.

Line 71, “When HPV genome is integrated, the virus main oncoproteins E6 and E7 induce deregulations in host cell cycle in order to promote its progression and proliferation. “ This sentence is misleading since HPV E6 and E7 deregulate p53 and pRb pathways regardless of the integrated or episomal state of the HPV genome. It was theorized in early studies of HPV that integration of the HPV genome resulted in increased expression of E6 and E7 and in turn cancer progression, but data is still needed that would support this theory.   Also, it’s not clear to what ‘its progression and proliferation’ refers to. Presumably, it refers to host cells but ‘host cells’ are not mentioned in the sentence.

We totally agree with your comment about pathways deregulation regardless of the integrated or episomal state of HPV. Besides, it was demonstrated that cervical cancer can arise from cells containing exclusively episomes. For HPV16, around 30% (or more) of cervical cancers develop in this way (Vinokurova S et al. Cancer Res 2008; Matsukura T et al. Virology 1989).

The paragraph was clarified and updated at lines 90 to 100.

Line 78, “This overexpression of p16 is a well established biological marker of HPV infection and is now referenced as a gold standard of active HPV infection.” High p16 protein expression is a marker for HNSCC being HPV positive not for active HPV infections.

The status of p16 as a marker of active infection is still controversial since Lechner et al. recently postulated that depending on E7 activity, p16 upregulation is now accepted as a marker of active HPV involvement in tumor cells.

However, we removed this sentence and developed a new paragraph regarding p16 overexpression at lines 106 to 122.

Line 80, “HPV/p16+ tumors have a transcriptionally active HPV infection where the virus is in its integrated form and induces the transcription of its genes while HPV/p16 - tumors have a latent HPV infection meaning there is no transcriptional activity for HPV genes.” This sentence has many problems.  All or virtually all HPV positive HNSCC cells have transcriptionally active HPV genomes whether p16 positive or not or independently whether episomal or integrated. It makes no sense to say episomal HPV positive cells have no transcriptionally active HPV genes when it’s established that expression of E6 and E7 is driving the cancer and E1 and E2 expression is required for the autonomous replication of the HPV episome. In fact, most studies that address this topic show that integration of the HPV genome into the host genome disrupts many HPV genes from being expressed as a result of splitting a gene and or dislocation of the primary HPV promoter from the E1, E2, E4 and E5 genes. While studies have shown a correlation between episomal/integrated states of HPV with the level of p16 expression, p16 overexpression has been used as a marker regardless of the HPV genome state though it’s significantly less than 100% accurate.

As explained above, we clarified our opinion about the latent infection, so we modified this sentence at lines 117 to 122 and we wrote a new paragraph regarding p16 overexpression.

There are some grammar and word-use issues, some of which are listed below. However, this list is not comprehensive and even the Abstract has at least one issue.

Line 49, “Thus, some clinical studies suggested that patients with a history of smoking and drinking abuses but without HPV infection would have a poorer overall survival than those with HPV- positive tumor and no history of alcohol and tobacco intoxication, and intermediate overall survival in patients with alcohol and smoke abuse and HPV- positive tumor.” This sentence is too messy to understand clearly.

Lines 65 to 70: we completely changed this sentence for a better understanding

Line 53, the term “would” makes the sentence awkward; “could” works better.

Line 74: the change was made as requested

Line 75, the term “binded” should be “bound”.

Line 102: the change was made as requested

Line 281, the term “researches” is uncommon and awkward; “research studies” is appropriate.

Line 377: the change was made as requested

Line 286, “Two studies interested to the expression of some important ligands by Treg, according to the HPV status.” This sentence is awkward and incomplete and needs restating.

Lines 387 to 388: The sentence has been reformulated

Line 287, the term “would” is awkward in this sentence.

Line 383: The sentence has been reformulated so that the term “would” was removed.

Line 433, “type” should be “types”.

Line 534: The error has been corrected

Line 481, the word “rational” does not make sense; the authors may have wanted to use “rationale”.

Line 635: The error has been corrected